# Risk Assessment of Chlorogenic and Isochlorogenic Acids in Coffee By-Products

**DOI:** 10.3390/molecules28145540

**Published:** 2023-07-20

**Authors:** Sascha Behne, Heike Franke, Steffen Schwarz, Dirk W. Lachenmeier

**Affiliations:** 1Postgraduate Study of Toxicology and Environmental Protection, Rudolf-Boehm-Institut für Pharmakologie und Toxikologie, Universität Leipzig, Härtelstrasse 16-18, 04107 Leipzig, Germany; sascha.behne@studserv.uni-leipzig.de (S.B.); heike.franke@medizin.uni-leipzig.de (H.F.); 2Fachbereich II (Fachgruppe Chemie), Berliner Hochschule für Technik (BHT), Luxemburger Strasse 10, 13353 Berlin, Germany; 3Chemisches und Veterinäruntersuchungsamt (CVUA) Karlsruhe, Weissenburger Strasse 3, 76187 Karlsruhe, Germany; 4Coffee Consulate, Hans-Thoma-Strasse 20, 68163 Mannheim, Germany; schwarz@coffee-consulate.com

**Keywords:** novel foods, coffee by-products, chlorogenic acid, CQA, risk assessment, caffeic acid, quinic acid, caffeoylquinic acid, dicaffeoylquinic acid, toxicology

## Abstract

Chlorogenic and isochlorogenic acids are naturally occurring antioxidant dietary polyphenolic compounds found in high concentrations in plants, fruits, vegetables, coffee, and coffee by-products. The objective of this review was to assess the potential health risks associated with the oral consumption of coffee by-products containing chlorogenic and isochlorogenic acids, considering both acute and chronic exposure. An electronic literature search was conducted, revealing that 5-caffeoylquinic acid (5-CQA) and 3,5-dicaffeoylquinic acid (3,5-DCQA) are the major chlorogenic acids found in coffee by-products. Toxicological, pharmacokinetic, and clinical data from animal and human studies were available for the assessment, which indicated no significant evidence of toxic or adverse effects following acute oral exposure. The current state of knowledge suggests that long-term exposure to chlorogenic and isochlorogenic acids by daily consumption does not appear to pose a risk to human health when observed at doses within the normal range of dietary exposure. As a result, the intake of CQAs from coffee by-products can be considered reasonably safe.

## 1. Introduction

Coffee is one of the best-selling and most consumed beverages worldwide [1]. In Germany, the coffee consumption per person is four cups a day, which corresponds to 6.7 kg of coffee beans per year [2]. There are several thousand varieties of the *Coffea* genus, with *Coffea arabica* and *Coffea canephora* being the most important species. Coffee is particularly cultivated and produced in tropical and subtropical regions along the equator (the so-called “coffee belt”), where ideal growth is possible due to the constantly warm temperatures and humid climate without extreme weather fluctuations. The most important coffee producers include Brazil, Vietnam, Colombia, Indonesia, Ethiopia, and India [2,3,4]. In addition to the main ingredient, caffeine (**1a**), the purine alkaloids theophylline (**1b**) and theobromine (**1c**), the diterpenes kahweol (**2**), cafestol (**3a**), and 16-*O*-methylcafestol (**3b**), as well as the flavonoid epigallocatechin gallate (**4**) and the polyphenolic chlorogenic acids (**5**) are present in coffee; these are shown in Figure 1 [4,5,6].

Due to the global climate crisis, the weather has changed to more extreme temperatures and less rainfall all over the planet. Hotter temperatures and extended dry periods in summer and extreme unusual frosts in winter have caused enormous coffee harvest losses over the last few years, which have also increased the price of coffee worldwide. Attempts are now being made to use all components of the coffee plant (the so-called “coffee by-products”) in addition to the coffee bean itself to increase sustainable coffee production and to reduce its carbon footprint.

Furthermore, the process of coffee production generates much waste (in the form of by-products) and wastewater. Especially with the many washing steps during production, large amounts of contaminated water with high carbon loading accumulate, which in turn represents a high environmental burden [7,8]. One of the most promising options to reduce the environmental impact of coffee production and make the process more sustainable is to utilize the resulting by-products by using the biologically active substances; for example, as functional food, novel food ingredients, or food supplements [8,9].

Before coffee by-products can be marketed in the food sector within the European Union (EU), it is of great importance that they acquire approval as novel foods. Novel foods are foods and/or food ingredients that are relatively new to the European market and therefore have no history of use as safe foods for human consumption [10]. The European legal framework for novel foods intends to protect human health and consumer interests. Regulation (EU) No. 2015/2283 defines “novel food” as any food that was not used for human consumption to a significant extent within the Union before 15 May 1997 [11,12]. Both the classification as traditional food from non-EU countries and the full application for authorization as a novel food require data of sufficient quantity and quality, including a description of the novel food, its manufacture, chemical analysis methods, and analytical and toxicological data to demonstrate that there is no safety risk to human health. The European Commission is responsible for the authorization of novel foods, which is often supported by the European Food Safety Authority (EFSA) since the EFSA carries out toxicological risk assessments to ensure (food) safety. Common coffee by-products, defined as any product derived from coffee production other than roasted coffee, are coffee flowers (blossoms), leaves, coffee cherry materials (husks, cascara, dried or fresh coffee cherries, and coffee pulp or mucilage), silver skin, parchment, green unroasted beans, and spent coffee grounds (Figure 2) [10,13,14,15]. This review deals with the fundamental aspects of the substance class of chlorogenic acids—their structural diversity, natural occurrence, and biological activities—and provides an insight into biosynthetic and totally synthetic approaches. This review also highlights the concentrations of chlorogenic acids in known coffee by-products and whether this poses a risk to human health and whether the consumption of these coffee by-products is safe and toxicologically harmless. In addition, a recommendation for the maximum daily intake in relation to chlorogenic acids in coffee by-products is suggested.

## 2. Literature Research

For this review, electronic searches of the literature were conducted, including of the databases PubMed (National Library of Medicine, Bethesda, MD, USA) and Google Scholar (Google LLC, Mountain View, CA, USA). A broad scope of search terms (and combinations thereof) was used, including coffee by-products, chlorogenic acid, caffeoylquinic acid, and toxicology of chlorogenic acid. In addition, standard works of literature were used to provide knowledge of the historical, botanical, biochemical, and synthetic background. Furthermore, databases such as the US Department of Agriculture (USDA) database as well as the EFSA homepage were searched for terms such as chlorogenic acid, caffeoylquinic acid and dicaffeoylquinic acid. No specific time restrictions were imposed for the selection of the literature in this review, as the rationale was to include a comprehensive overview and because much of the available literature on the topic comprises older publications.

## 3. Chlorogenic Acids

The quinic acid derivatives shown in this review conform to the IUPAC nomenclature. Unfortunately, there are many publications depicting structures that are not numbered according to the IUPAC system, or that do not specify the applied numbering system. The authors have diligently aligned the nomenclature in all reviewed papers to conform with IUPAC usage.

### 3.1. Structures, Properties, and Natural Occurrence

Originally, the naturally occurring esters of *trans*-configurated caffeic acid (**6**) and (–)-quinic acid (**7**) were known by the trivial name chlorogenic acid (CQA, singular). In 1846, Payen used the term “chlorogen acid” for the first time and isolated the crystalline potassium caffeine chlorogenate complex from green coffee (*Coffea arabica*) [16,17]. The name chlorogenic acid comes from the Greek χλωρός (*khloros*, light green) and γένος (*ghenos*, producing) and refers to the intense green color that results when chlorogenic acids are oxidized [2]. The origin of the name may also be due to a green color that appears when chlorogenic acid is treated with an aqueous solution of ammonia in the presence of atmospheric oxygen [18]. The prefix “chloro” is misleading in that the substances have no chlorine atoms in their chemical structure. The first time that pure chlorogenic acid was obtained in crystalline form with a melting point of 206–207 °C was in 1907, with evidence being provided by the formation of caffeic acid and quinic acid after alkaline hydrolysis [19]. Twenty-five years later, Freudenberg confirmed that chlorogenic acid is a caffeic acid–quinic acid conjugate [20].

Today, several structurally similar compounds belong to the class of chlorogenic acids (CQAs, plural), which also includes esters of other hydroxycinnamic acids such as ferulic acid (**8**) and *p*-coumaric acid (**9**) [21,22,23,24,25]. Chlorogenic acids are secondary metabolites and belong to the biologically active dietary polyphenols with a phenylpropanoid moiety. They are found in numerous plants, fruits, and vegetables [26] and play an important role as an intermediate in the biosynthesis of lignin [27,28]. Figure 3 shows an overview of the chemical structures of the main chlorogenic acids (**10**–**13**).

It was not until 1932 that Fischer and Dangschat suggested that the substance isolated by Freudenberg must be 3-caffeoylquinic acid (3-CQA, **10a**) [29]. According to today’s IUPAC nomenclature, the 3-CQA postulated by Fischer and Dangschat is 5-caffeoylquinic acid (5-CQA, **11a**). Various publications on further structural analogues followed in the subsequent years.

To date, over 80 different chlorogenic acids have been identified exclusively in green coffee [30], and more than 400 CQAs are currently known [31]. The structural diversity of chlorogenic acids results from the fact that the hydroxycarboxylic acid has four hydroxy groups that are arranged differently in space. According to the IUPAC nomenclature, the two hydroxy groups at C3 and C4 of (–)-quinic acid (**7**) are arranged equatorially, while the OH group at position 5 is axial (see Figure 4) [32,33]. Each of these OH functions of (–)-quinic acid can form corresponding esters with the hydroxycinnamic acid derivatives **6**, **8**, and **9**. In addition to the chlorogenic acids shown in Figure 3, other hydroxycinnamic acid–quinic acid conjugates are also known, e.g., those made from sinapinic acid or 3,4,5-trimethoxycinnamic acid and (–)-quinic acid (**7**) or multiple mixed esters such as caffeoyl-feruloylquinic acids (CFQAs) [7,15,34].

Among the configurational/conformational/regio-isomers 3-CQA (**10a**), 4-CQA (**12a**), 5-CQA (**11a**) and 1-CQA (**13**), 5-CQA (**11a**) is the most prevalent chlorogenic acid in coffee [15,35]. The epimer 1-CQA (**13**) occurs in nature very rarely and in very small amounts. Parejo et al. were able to identify 1-CQA in fennel (*Foeniculum vulgare*) using LC-DAD-ESI-MS/MS as the chromatographic method [36].

In addition, the chlorogenic acids can be converted into one another by transesterification (Figure 5). The solvent used in the extraction process also has an impact on the transesterification reaction. For example, the chlorogenic acid 1-caffeoylquinic acid (**13**), which tends to be underrepresented in plants, can be formed via acyl migration being accelerated by an increased amount of water in the organic extractant or in the plant material [37,38,39]. Transesterification reactions are also possible with FGAs and *p*-CoGAs in order to obtain the other structural isomers.

In addition to caffeine, chlorogenic acids occur as an ingredient in green and roasted coffee [40]. However, the content of chlorogenic acids in green coffee is greater than in roasted coffee [41,42,43,44]. During the roasting process, CQAs can be converted by dehydration into chlorogenic acid lactones such as 3-*O*-caffeoylquinic-1,5-γ-lactone (3-CGL, **14**) and 5-*O*-caffeoylshikimic acid (dactylifric acid, **15**), which results in a decreased amount of classic CQAs [29,45,46]. Figure 6 shows examples of possible roasted products based on monocaffeoylquinic acids.

The monocaffeoylquinic acids can be found in numerous plants, such as valerian (*Valeriana officinalis*) [47], sunflower (*Helianthus annuus*) [48], bamboo (*Phyllo-stachys edulis*) [49], heather (*Calluna vugaris* from *Ericaceae*) [50], lemon balm (*Melissa officinalis*) [51], nettle (*Urtica dioica*) [52], and Japanese honeysuckle (*Lonicera japonica*) [53], but also in the pulp of blueberries (*Vaccinium corymbosum*) [54], apples (*Malus domestica*) [55], grapes (*Vitis vinifera*) [56], eggplant (*Solanum melongena*) [57], peaches (*Prunus persica*) [58] and dried plums (*Prunus domestica*) [59], and in the roots of chicory (*Cichorium intybus*) [60]. Chlorogenic acid (**11a**), cryptochlorogenic acid (**12a**), and neochlorogenic acid (**10a**) can also be detected in the leaves of the African mallow (*Hibiscus sabdariffa* from *Malvaceae*) [61] and in walnuts (*Juglans regia*) [62].

Structures having more than one caffeic acid residue are called dicaffeoylquinic acids or isochlorogenic acids and can be found, e.g., in coffee and in plants of the family *Asteraceae* [63]. Prominent isochlorogenic acids are shown in Figure 7. The most abundant isochlorogenic acid in extracts from Indian pennywort (*Centella asiatica* from *Apiaceae*) and other plant sources is 3,5-DCQA (**16a**) [64,65]. In 1954, the dicaffeoylquinic acid 1,3-DCQA (**16b**) was isolated and characterized as the first 1-acyl quinic acid from the artichoke (*Cyanara cardunculus*) [21,66]. Also called cynarine, 1,3-DCQA (**16b**) has antioxidant and anticholinergic effects. Other isochlorogenic acids have also been found in the leaves of sweet potato (*Ipomoea batatas*) and white and green tea (*Camellia sinensis*) [67].

Chlorogenic and isochlorogenic acids have demonstrated various effects in several studies. CQAs have antioxidant [68], antibacterial [69], antiviral [70], antidiabetic [71], neuroprotective [72,73], anti-inflammatory [74], and cytostatic effects [75,76]. CQAs have been used therapeutically in some clinical treatments as well, e.g., in the treatment of cardiovascular diseases [77] and arterial hypertension (high blood pressure) [78]. The broad scope of bioactivities and pharmacological applications have attracted much attention from research scientists. Novel synthetic chlorogenic-acid amide analogues have shown both higher chemical stability compared with classic chlorogenic acids and biological activity against the hepatitis C virus [79]. Several other chlorogenic acid derivatives have also been made synthetically accessible and have other promising properties; for example, they have an antifungal effect and effectively inhibit the HIV integrase/protease [80].

In addition to chlorogenic acid and isochlorogenic acid, there are other quinic acid conjugates with a higher number of caffeoyl residues. Figure 7 shows the chemical structures of known tri- and tetracaffeoylquinic acids. The ubiquitous occurrence of tricaffeoylquinic acids (TCQAs) in the plant kingdom has been known for a long time. Bates et al. characterized and identified 3,4,5-TCQA (**17a**) for the first time in 1983 from methanolic extracts of black-banded rabbitbrush (*Chrysothamnus paniculatus* of the *Compositae* family) [81].

In 1993, Agata et al. detected 1,3,5-TCQA (**17b**) in the fruits of the common cocklebur (*Xanthium strumarium*) [82]. A year earlier, Merfort reported the discovery of 1,4,5-TCQA (**17c**) from the flowers of *Arnica montana* and *Arnica chamissonis* by extraction with ethyl acetate [83]. In 2018, Liu et al. demonstrated the occurrence of 1,3,4-TCQA (**17d**) in *Duhaldea nervosa* chromatographically and spectroscopically [84]. Since then, the number of tricaffeoylquinic acids in different plants with a variety of biological activities has increased significantly. Kim et al. reported the interesting observation that sunlight significantly increases the amount of CQAs when isolating and quantifying 1,3,4-TCQA and 1,3,5-TCQA from *Ligulara fischeri* [85]. The antihyperglycemic effect of 3,4,5-TCQA was confirmed in a study with a water-soluble fraction of Brazilian *propolis* [86]. In 1994, Scholz et al. discovered 1,3,4,5-*O*-tetracaffeoylquinic acid (**18**) from the aerial parts of *Pluchea symphytfolia* [87], which showed significant inhibition on the growth of bacterial strains such as *Escherichia coli*, *Bacillus subtilis*, and *Micrococcus luteus* and had a weak anthelmintic effect in in vitro studies [88]. In addition to the TetraCQA (**18**) isolated by Scholz et al., cyclobutane structural analogues have also been found, which formally result from a pericyclic reaction via the [2+2]-cycloaddition between two neighboring caffeoyl residues [60]. It is particularly impressive that 1,3,4,5-*O*-tetracaffeoylquinic acid (**18**) inhibits *Trypanosoma brucei* RTPase Cet1 (TbCet1) extraordinarily effectively but is, on the other hand, necessary for the proliferation of procyclic cells [89]. Using a colorimetric high-throughput screening (HTS) assay, the inhibitory activity (IC_50_) of TetraCQA was found to be in the submicromolar range at 13 nM and is the most effective inhibitor compared with around 20 other TbCet1 inhibitors. Isochlorogenic acid A (**16a**) is also effective at inhibiting TbCet1, with an IC_50_ of 70 nM. The number of naturally occurring derivatives of the CQA family could be determined as follows: DCQA > TCQA > TetraCQA [7].

### 3.2. Biosynthetic Pathways and Totally Synthetic Approaches

Plants can biosynthesize chlorogenic acids through a combination of the shikimic acid pathway and the phenylpropanoid pathway (Figure 8), with CQAs being an important intermediate in the biosynthesis of lignin [16,81,90]. The shikimic acid pathway starts with the enzyme-catalyzed cyclization of phosphoenolpyruvate (PEP, from glycolysis) and erythrose 4-phosphate (E4P, from the pentose phosphate pathway), in which 3-dehydroquinic acid is formed via 3-deoxyarabinoheptulosanate-7-phosphate (DAHP). Following the dehydratase-catalyzed elimination of water, 3-dehydroshikimic acid is then formed, which reacts by redox reaction under the influence of NADPH to form shikimic acid. The key intermediate, shikimic acid, is then converted into chorismic acid by ATP-dependent kinase-catalyzed phosphorylation and subsequent phosphate elimination. The cyclohexadiene prephenic acid is then formed by the enzymatically catalyzed Claisen rearrangement, which, after decarboxylation and subsequent transamination, results in the product of the shikimic pathway, L-phenylalanine. L-phenylalanine is the starting point of the phenylpropanoid pathway [91]. In the first step, ammonia is split off via phenylalanine ammonia lyase (PAL), resulting in cinnamic acid as a reaction product. Then, via *p*-coumaric acid, cinnamate 4-hydroxylase (C4H) and 4-hydroxycinnamoyl-CoA ligase (4CL) generate caffeic acid as another important key intermediate in the biosynthetic pathway of CQAs. Starting from caffeic acid (**6**), for example, 5-CQA (**11a**) can be formed in two different ways: (a) by 4CL via caffeoyl-CoA with a subsequent transferase-catalyzed (HCT/HQT) reaction and (b) by UGT84 via caffeoyl glucoside with a subsequent hydroxycinnamoyl *D*-glucose: quinate hydroxycinnamoyl transferase (HCGQT)-catalyzed reaction [92,93]. Also known is the synthesis of 5-CQA by (c) the enzymatic reaction of *p*-coumaric acid (9) to *p*-coumaroyl-CoA and the subsequent formation of *p*-coumaroylquinic acid. A subsequent reaction with the enzyme coumarate 3-hydroxylase (C3H) then allows biosynthetic access to 5-CQA (**11a**) [94]. How plants produce DCQAs or even TCQAs biosynthetically has not yet been fully elucidated, but it is likely that the acylation of monocaffeoylquinic acids with caffeoyl-CoA is catalyzed by HCT/HQT, which has already been demonstrated in tomatoes and sweet potatoes [95,96].

However, since plants always contain mixtures of many secondary metabolites, the isolation of CQAs from the corresponding plant extracts involves a great deal of effort. When extracting CQAs from plant material, chlorogenic acids can undergo various degradation reactions and/or isomerization under the influence of temperature, pH, and light, so the original CQA ratio can be strongly influenced and even falsified [97]. In addition to the storage conditions of plant materials, problems are also caused by thermal process steps that have an impact on the amount and ratio of chlorogenic acids. As mentioned before, the proportion of chlorogenic acids in thermally treated, and thus roasted, coffee is significantly reduced compared with green coffee [98]. Chemical conversion processes such as epimerization, acyl migration, and dehydration are responsible for this [99]. In order to adequately investigate the diverse biological activities of CQAs, chemical syntheses for the preparation of larger amounts of CQAs in pure form have come into focus. To date, there are only a few totally synthetic strategies starting from differently protected quinic acid and caffeic acid derivatives with moderate to good yields. An overview of possible synthesis strategies for 5-CQA (**11a**) starting from quinic acid acetals is shown in Figure 9 [100,101,102]. In 1955, Panizzi et al. reported the first synthesis of 5-CQA (**11a**). However, the synthesis from quinic acid was only possible in seven steps, with a low yield of <5% [103,104]. The choice of the protecting group and its removal as quantitatively as possible in the final step of the synthesis proved to be a major challenge. During the synthesis, it is important to keep in mind that deprotection under basic conditions should be avoided, since CQAs have been shown to be sensitive to oxidation reactions. Synthetic strategies should therefore exclusively rely on protecting groups that can be removed under acidic conditions. Hemmerle et al. were able to produce 5-CQA in a 5-step synthesis with an overall yield of 20–32%, starting from a quinic acid acetal and using acid-labile-protecting groups [91]. Sefkow was the first to be able to increase the yield in the synthesis of 5-CQA in 2001 to 65%, when he presented a 4-step synthesis strategy starting from quinic acid (**7**) that proceeded via a fully silyl-protected isolable intermediate and that also used acid-labile-protecting groups [92]. Fourteen years later, Kadidae et al. reported a short total synthesis of 5-CQA, but this synthetic approach was not quite as successful (with 35% over four steps) as that of Sefkow [93]. From the point of view of sustainability, the synthesis route of Hemmerle et al. was the best, as predominantly inexpensive and stable reagents were used. To date, however, the synthesis of 5-CQA according to Sefkow’s synthetic protocol is the most promising route, with the fewest number of synthetic steps and the highest yield. Sefkow et al. also reported synthetic strategies for the synthesis of 1-CQA, 3-CQA, and 4-CQA via quinic acid acetals with differently protected caffeic acid derivatives [105]. In addition, synthesis routes for the preparation of isochlorogenic acids are also known in the literature; however, these synthesis routes have not yet been able to surpass the synthesis protocol of Sefkow, and they are only associated with low to moderate yields [106,107,108,109].

The synthesis of tri-*O*-caffeoylquinic acids (TCQAs) from acetal-protected quinic acid lactones by acylation has been successfully demonstrated several times using a variant of the Schotten–Baumann method and via Steglich esterification [81,110].

## 4. Amounts of CQA in Coffee By-Products

As already mentioned, chlorogenic acids are found in a variety of plants. In coffee, the level of CQAs can vary widely depending on the degree of roasting and the geographical origin. The CQA content in roasted coffee ranges from 0.4–3.4 g/100 g (*C. arabica*) and 2.1–6.4 g/100 g (*C. canephora*) [44,111,112,113], while the amount of 5-CQA in green coffee has been shown to be higher at 6.0–9.0 g/100 g [114,115]. Processing, especially roasting, modifies dramatically the phenolic composition of coffee, but creates the characteristic aroma, flavor, and color of coffee beverages. In comparison with coffee and *Prunus* spp. cherry juice (85 mg/L), parts of the potato plant (leaves, sprout, root material) and the uncooked potato tuber contain very little amounts of 5-CQA, about 0.02–0.75 g/100 g [116,117]. Table 1 shows the amounts of monocaffeoylquinic acids and dicaffeoylquinic acids (isochlorogenic acids A–C) from coffee by-products with reference to the literature. It is noticeable that the respective values for CQAs in all coffee by-products vary quite widely. This is due to a number of factors, such as species and variety, different post-harvest processing methods, degree of ripeness, extent of environmental conditions, agricultural practices, and region of origin. What is also striking is that 5-caffeoylquinic acid (**11a**) is the most frequently occurring monocaffeoylquinic acid in all coffee by-products in terms of quantity, and the amounts of 3-CQA and 4-CQA are comparatively low. For this reason, the total amount of polyphenolic compounds is often given in the literature, with 5-CQA making up the largest proportion. Based on the amount of 5-CQA, the following sequence of coffee by-products can be formulated according to the amounts in Table 1: coffee husks > coffee pulp > silver skin > spent ground coffee > green unroasted beans > coffee leaves > coffee flowers (blossoms) > parchment. From the dicaffeoylquinic acids, 3,5-DCQA (**16a**) is mostly found in coffee by-products, although not all coffee by-products have been shown to contain DCQA in the literature.

Wirz et al. determined a maximum content of 5-CQA in coffee flowers (blossoms) of 2.64 g/100 g, whereas the amounts of 3-CQA and 5-CQA were negligibly small or little investigated. A concentration of up to 5.84 g/100 g 3,5-DCQA and only 0.25 g/100 g 3,4-DCQA was found in coffee flowers. The concentration of 3,5-DCQA is about twice as high as the concentration of 5-CQA. A similarly high concentration of up to 5.79 g/100 g DCQAs was also found in spent coffee grounds. Even if the information relates to the total amount of all dicaffeoylquinic acids, it can be assumed that 3,5-DCQA (isochlorogenic acid A) is the main isochlorogenic acid in terms of quantity. Using HPLC, a concentration of up to 5.58 g/100 g 5-caffeoylquinic acid could be determined from coffee leaves [118]. The content of 3-CQA and 4-CQA and especially the concentration of isochlorogenic acids were extremely low in comparison. Rodriguez-Gomez et al. determined the concentration of 5-CQA in coffee leaves using LC-EC and LC-qToF-MS to be 1.0–190 mg/L [121], with the different concentrations often varying greatly for reasons already explained. At 0.13–2.44 g/100 g, 5-CQA is also the monocaffeoylquinic acid in coffee pulp with the highest concentration. According to Esquivel et al., 3,4- and 3,5-DCQA are represented 2 to 40 times less abundantly than 5-CQA at 0.06 g/100 g [125]. In contrast, in coffee husks, Palomino Garcia et al. found a 5-CQA concentration of 13.3 g/100 g. A similarly high concentration of 5-CQA (13.4 g/100 g) was found by Regazzoni et al. in green unroasted coffee beans using HPLC-UV for quantification.

In silver skin, the concentration of 5-caffeoylquinic acid (5-CQA) determined from the literature is between 0.02 (minimum) and 9.0 g/100 g (maximum). Only Regazzoni and co-workers were able to extract and quantify the other mono- and dicaffeoylquinic acids by liquid chromatography [131]. In comparison with the other coffee by-products, the determined concentrations of 3-CQA (up to 5.16 g/100 g) and 4-CQA (up to 6.02 g/100 g) come very close to the concentration of the main chlorogenic acid 5-CQA (up to 8.89 g/100 g), while in other by-products 3-CQA and the 4-CQA occur in low concentrations compared with 5-CQA. The concentration of dicaffeoylquinic acids in silver skin, on the other hand, is largely equally distributed among the three DCQAs in a ratio of about 1:1:1 (each with up to 1.00 g/100 g) [131]. On the contrary, the concentration of 5-CQA in the coffee by-products parchment and spent coffee grounds is quite low at 0.4–0.61 g/100 g and 1.32–2.30 g/100 g, respectively [127,129,131,136]. However, it is interesting that the monocaffeoylquinic acids 3-CQA, 4-CQA, and 5-CQA in spent coffee grounds occur in a ratio of about 1:1.1:1.3 and differ less in terms of quantity than in other coffee by-products [131].

According to Murthy and Naidu, coffee by-products in total contain about 2.3–3.0% chlorogenic acids as the phenolic compound [137]. To verify this statement, the concentrations from the literature (from Table 1) for each coffee by-product were listed using the unit g/100 g, which allows the direct conversion into a percentage (%). Table 2 shows the results. The literature concentrations result in a concentration range of 0.8–5.5% (min–max) chlorogenic acids, with a total average value of 3.1%. The determined range is a bit wider than given by the authors, but the calculated total average value fits very well.

For comparison, Belitz et al. reported that green coffee from *C. arabica* contains 3.0–5.6% and *C. canephora* 4.4–6.6% chlorogenic acid (based on dry matter), while the chlorogenic acid content in roasted coffee is 2.7% for *C. arabica* and 3.1% for *C. canephora*, based on the dry matter [138]. Perrone et al. describe a chlorogenic acid content of 4–12% in the raw coffee components in the mass [139], which can be attributed to the transformation into related compounds (see Figure 6).

## 5. Absorption, Distribution, Metabolism, and Excretion

How a substance is absorbed in the body, distributed, metabolized, and then eliminated (ADME) depends largely (but not exclusively) on the physical and chemical properties of the substance. Important physical and chemical properties of 5-CQA and 3,5-DCQA are shown in Table 3. The pharmacokinetics of phytochemical hydroxycinnamic acids (including chlorogenic acids) have been extensively studied over the last few years. In terms of quantity, 5-CQA and 3,5-DCQA are the most frequently occurring chlorogenic acids in coffee by-products, which is why only these two representatives are considered in more detail below.

The positive health effects triggered by chlorogenic acids after oral intake correlate with their bioavailability in the body. The overall oral bioavailability depends on certain factors, including water solubility and (chemical) stability but also physicochemical properties (summarized in Table 3). Water solubility plays a major role, since substances must be dissolved in order to be absorbed in the gastrointestinal tract [143]. According to Horter et al., chlorogenic acid as well as hydroxycinnamic acids such as caffeic acid, ferulic acid, rosmarinic acid, and *p*-coumaric acid have a water solubility of >0.1 mg/mL, characterizing them as water-soluble compounds [141].

In all gastrointestinal assays, it could be shown that CQAs have a stability of 48% [144]. A study by Ren et al. showed that the gastric and intestinal phases chemically affected the stability of CQAs [145]. The stability is due to the structure of CQAs: the ester function is a reactive part and a good target for nucleophilic substrates such as amino acids, peptides, and proteins and can lead to secondary products. Furthermore, it is already known that CQAs undergo pH-dependent isomerization reactions. It was shown that 5-CQA is converted into the isomer in rats, human plasma, and phosphate buffer at 37 °C and a pH of 7.4 [146].

In in vitro studies, the intracellular accumulation of hydroxycinnamic acids was determined to be <2%, and the rate of transport, defined as the amount permeating toward the basolateral membrane, for 5-CQA was 0.1–0.3%. In comparison with other prominent hydroxycinnamic acids such as caffeic acid, coumaric acid, and rosmarinic acid, the rate of transport of CQAs was the lowest [147]. In the acidic pH of gastric cells (pH = 3.0), 50% of 5-CQA exists in its uncharged form, whereas the remaining 50% negatively charged, exists in Caco-2 cells (pH = 7.4) [143,148].

The metabolism of 5-CQA is formally divided into phase 1 (functionalization) and phase 2 (conjugation), and the enzymes mainly involved and responsible for the degradation of drugs are the cytochrome P450 isoenzymes (CYP), catechol-*O*-methyltransferase (COMT), sulfotransferases (SULTs), and UDP-glucuronosyltransferases (UGTs). These enzymes metabolize 5-CQA by isomerization, hydrolysis (forming caffeic acid), methylation (forming 5-feruloylquinic acid, 5-FQA), glucuronidation (forming CQA glucuronide), and sulfonation. The microbial metabolism in the intestinal tract with human fecal microbiota revealed that 5-CQA was undetectable over a time interval of 0.5–2.0 h and that 3-hydroxyphenylpropionic acid (3-HPPA) was identified as a metabolite [149]. Ludwig et al. reported in an in vitro study that after 6 h incubation of 5-CQA with human fecal microbiota, 11 metabolites were formed, with 3-HPPA, dihydroxycaffeic acid, and dihydroxyferulic acid making up 75–83% [150]. Additionally, it is known from ex vivo studies that the amount of 5-CQA absorbed in the porcine jejunal segment is less than 1.5% and that 5-CQA is more permeable in the duodenum than in the ileum, jejunum, and colon [151]. In an in situ study by Lafay et al., when 5-CQA was administered into the stomach of rats, 9.4% of 5-CQA was recovered in the gastric vein and 4.6% of 5-CQA was recovered in the aorta in its intact form, with no evidence of metabolite formation [152].

In order to obtain further pharmacokinetic data and in particular, information on distribution, experiments were carried out on Wistar rats, Sprague-Dawley rats, and Kunming mice with oral doses of 1–1200 mg/kg body weight (bw) [143]. At an oral dose of 50 mg/kg bw in Wistar rats, a volume of distribution (V_D_) of 97.5 L/kg and a clearance (CL) of 39.0 L/h/kg could be determined. The low V_D_ value of 5-CQA also indicates a tissue distribution since the values were superior to the real animal body volume [153]. In low doses of 1.0–8.0 mg/kg in Sprague-Dawley rats, the clearance was determined at the maximum time of T_max_ = 0.25–1.50 h to be 0.62–0.73 L/h/kg [154]. Considering the physicochemical characteristics of hydroxycinnamic acids, ferulic acid is the most lipophilic compound with a plasma protein-binding value of 73.5%, while 5-caffeoylquinic acid has a plasma protein-binding value of only 25.6% [155]. According to studies by De Oliveira et al., specific tissues reached by 5-CQA were the kidney, liver, and muscle [156]. Quantitative tissue distribution analyses were carried out by Chen et al. with the result that 5-CQA was distributed in the body in the following order: liver > kidney > heart > spleen > lung (based on the observed area under the curve, AUC) [157]. The excretion rate of 5-CQA (free form) in the urine over a period of 6–48 h was only 0.04% and is therefore negligible [158]. However, studies with a reduced time (8–24 h) showed even higher values than the 48 h study, with 0.07–0.50% for 5-CQA [159,160]. The 5-CQA renal clearance (CL = 0.15–0.29 L/h/kg) demonstrated that the hepatic extraction ratio was responsible for 23.1–28.2% of the 5-CQA elimination [154,161] and implied that 5-CQA underwent renal metabolism [143].

Not much is known about the pharmaco-/toxicokinetics of 3,5-dicaffeoylquinic acid compared with 5-CQA. Wang et al. reported in a study that 3,5-DCQA is metabolized to CQA and caffeic acid and excreted with a clearance of 1.07 L/h/kg [162]. The same group undertook in-depth investigations into the excretion of 3,5-DCQA in an in vivo study with Sprague-Dawley rats, in which they collected urine at intervals of 0–24 h and measured the concentration of 3,5-DCQA using LC/ESI-MS [163]. Bile samples were collected at intervals of 0–2, 2–4, 4–6, 6–8, 8–10, 10–12, and 12–24 h after administration, and a concentration of 1.9 ± 0.8% was determined, with 1.6 ± 0.9% already being eliminated after 4 h (simultaneous analysis of CQA, CA, and DCQAs). Excretion at <10% is very low, and these results may indicate that some DCQA was converted to CQA and excreted in the urine, consistent with the literature results. The excretion phase half-life was within the range of 0.3–0.6 h, which suggests that phenolic acids were rapidly eliminated throughout the organism, and the volume of distribution V_D_ was 1.8–3.9 L/kg.

Mehta et al. examined the V_D_ after the oral administration of *A. fragrans* to rats at a dose of 0.16 g/kg (n = 6) and determined it to be 4.29 × 10^5^ mL/kg [164]. DCQAs, determined using UHPLC-MS/MS, were mainly absorbed in the small intestine and their isomers were also absorbed quickly. In comparison, MCQAs were chiefly found in tissues, not in plasma, and DCQA as well as MCQA isomers were found in the ovary and uterus, while some could pass through the blood–brain barrier [165].

## 6. Toxicological Information

As already mentioned, human exposure occurs from the ingestion of medicinal or dietary plants containing chlorogenic and isochlorogenic acids. To come to an assessment of the toxicology of chlorogenic acids and isochlorogenic acids, the acute and chronic toxicity, but also the genotoxicity, cytotoxicity, and neurotoxicity as well as the mutagenic, teratogenic, and carcinogenic effects must be evaluated.

### 6.1. Acute and Subchronic Toxicity

Acute toxicity, also known as short-term toxicity, refers to the harmful effects that a substance can have on an organism when it is exposed to a high dose for a short period of time. Chronic toxicity refers to the long-term adverse effects of a substance on living organisms, typically resulting from repeated or continuous exposure over an extended period. Both are critical aspects of toxicological risk assessment, as they can provide valuable information on the potential hazards of a substance and help to establish safe exposure levels. There have been several studies conducted on the acute and subchronic toxicity of 5-CQA and 3,5-DCQA in animals.

In 1983, Schafer et al. found that the oral LD_50_ of chlorogenic acid in red-winged blackbirds (*Agelaius phoeniceus*) was greater than 100 mg/kg bodyweight (bw), equivalent to 0.28 mmol/kg, but did not report the sex and strain of the animal models [166]. In a study on female non-pregnant Wistar rats, a single-dose of 5-CQA at 2000 mg/kg was administered orally, and it was observed that the rats did not show any clinical signs of toxicity or mortality [167]. All test animals showed normal body weight gain at the end of the experiment, and the study concluded that the oral median lethal dose (LD_50_) of 5-CQA was greater than 2000 mg/kg in rats (p.o.). This suggests that 5-CQA has a low acute toxicity and is relatively safe when consumed in small amounts. In the same publication, the group also performed a 90-day subchronic toxicity study in compliance with the OECD guideline 408 with 100 healthy Wistar rats (50 male, 50 female). Here, the animals were orally administered 250, 500 and 1000 mg/kg of a CQA extract. The intake did not cause any toxic symptoms or abnormalities, but there were significant alterations in parameters such as food consumption, relative organ weight of the brain and spleen, and some biochemical parameters in comparison with the control group. These changes were toxicologically insignificant and within the physiological range.

Additionally, a study on mice showed similar results, with an LD_50_ of >2000 mg/kg. In that study, the maximum dosage was 5000 mg/kg body weight, which caused tonic convulsion followed by very rapid death [168]. Changes in body and organ weight are a clear indication of damage caused by the ingestion of a toxic substance. It was found that 5-CQA did not induce any significant changes in body weight, food consumption, or organ weights, indicating that it does not have any toxic effects. Furthermore, no adverse effects were shown in the subacute toxicity test with the oral administration of 1000 mg/kg (highest dose) over a period of 30 days. The highest dose of 1000 mg/kg bw therefore was taken as a no-observed-adverse-effect level (NOAEL) to estimate a human equivalent dose of 189 mg/kg bw per day or 13.2 g/day (for a 70 kg adult) [168].

In a sub-chronic toxicity study conducted in humans, doses of 330 mg CQAs dissolved in 100 mL of water were administered orally for 6 months [169]. The study used healthy people of both sexes, and the participants ingested one 100 mL bottle daily before bedtime. In terms of composition, the CQAs consisted of ~58% CQAs (total of 3-CQA, 4-CQA, and 5-CQA), ~22% DCQAs (total of 3,4-DCQA, 3,5-DCQA, and 4,5-DCQA) and ~20% feruloylquinic acids. The results showed that there was no significant toxicity induced by the ingestion of CQAs. Incidentally, the authors of the study were able to detect an improvement in cognitive function through repeated neurocognitive testing of the participants. Four other sub-chronic toxicity studies have been conducted in animals. In 1994, Kitts and Wijewickreme reported that the exposure of 9–12 female mice for 10 weeks to chlorogenic acid in the diet (0.2%, 2000 ppm) did not induce any clinical symptoms of toxicity, and the treatment did not affect body, liver, or intestinal weights [170]. Five male Sprague-Dawley rats exposed to 1% chlorogenic acid in the diet for 3 weeks showed reduced adrenal and kidney weights, while 2% chlorogenic acid in the diet for 4 weeks induced forestomach hypoplasia in a study with six male Fischer-344 rats [171,172]. Chaube et al. reported in 1976 that treatment with a daily injection (i.p.) of chlorogenic acid of up to 500 mg/kg per day in 9-week-old Wistar rats over a period of 8 days did not induce lethality [173].

Limited data are available on the chronic toxicity of CQA in humans, as most of the studies conducted so far have focused on its potential health benefits rather than its toxic effects. However, some human studies have provided insights into the safety of CQA when consumed over prolonged periods. In a randomized, controlled trial conducted in healthy human subjects, an instant coffee containing green-coffee extract with a high content of chlorogenic acid (90–100 mg of chlorogenic acids per 200 mg of green-coffee extract, with equal amounts of 3-CQA, 4-CQA and 5-CQA) was administered orally for 12 weeks [174]. The study found no significant adverse effects on body weight, blood pressure, heart rate, blood lipid levels, and liver or kidney function parameters. The study concluded that CQA supplementation was safe and well tolerated in healthy human subjects.

In another randomized, placebo-controlled human intervention study, green-coffee extract (200 mg of extract per capsule, containing 90–100 mg per capsule, with equal amounts of the MCQAs 3-CQA, 4-CQA and 5-CQA) vs. maltodextrin-placebo was provided to overweight and obese subjects [175]. Thirty participants consumed two capsules of green-coffee extract daily for 60 days. The study found no significant adverse effects of CQA intake on the risk of chronic diseases, and the authors concluded that CQA intake from coffee consumption was not associated with increased health risks. Overall, most animal studies on the chronic toxicity of CQA have shown that it has a low toxicity profile, with no significant adverse effects observed at doses within the normal range of dietary exposure.

In animal studies, acute oral toxicity of 3,5-DCQA has been investigated using different animal models and doses. For instance, a study by Simeonova et al. (2019) examined the acute oral toxicity of 3,5-DCQA in spontaneously hypertensive rats (SHRs) [142]. In this in vivo study, two phases were performed to determine acute oral toxicity. In the first phase, animals were treated orally with of 3,5-DCQA at 10, 100, and 1000 mg/kg bw. The surviving SHRs were observed for significant signs of toxicity and/or death (up to 24 h). In the second phase, nine SHRs were administered (p.o.) with higher doses (1600, 2900 and 5000 mg/kg, three SHRs for each concentration) of 3,5-DCQA. No mortality could be detected at 1600 mg/kg 3,5-DCQA but one of the three animals died (33% mortality) after a dose at 2900 mg/kg. At an acute dose of 5000 mg/kg, the mortality was 100%. According to Derelanko et al., therefore, 3,5-DCQA with an LD_50_ of 2154 mg/kg bw could be classified as “slightly toxic” in the acute oral toxicity test administered orally to SHRs [176].

Acute dermal toxicity refers to the adverse effects that occur after direct skin contact with a toxic substance. No study could be found on the acute dermal toxicity of 3,5-DCQA. However, a patent from Seo et al. (2021) describes the application of 3,5-DCQA as an active ingredient in a cosmetic or pharmaceutical composition for improving human skin barrier damage and/or alleviating skin inflammation [177]. This indicates that 3,5-DCQA has low acute dermal toxicity. Inhalation toxicity studies on 5-CQA and 3,5-DCQA are limited, as they are not commonly administered through inhalation routes. Most of the available toxicological data focused on acute oral toxicity, as these are the primary routes of exposure for humans and animals.

### 6.2. Genotoxicity and Mutagenicity

Various test methods with their respective advantages and disadvantages are available for testing genotoxicity/mutagenicity. Usually, the test battery starts with tests for adduct formation (by direct action on the DNA), and then experiments on microorganisms and mammalian cells follow before animal experiments are used (tier concept). The results from the literature are summarized in Table 4. To determine the genotoxicity of chlorogenic acid, in vitro experiments were carried out independently by different research groups on isolated DNA [178,179,180,181]. Chlorogenic acid (250 µM) was found to induce DNA double-strand breaks on isolated λ DNA in acellular systems. In the presence of copper(II) ions, DNA damage by chlorogenic acid could also be observed in phage DNA. Yoshie et al. showed that CQA did not lead to strand breaks in plasmid DNA at a concentration of 100 µM CQA; this could only be observed in combination with NO-releasing compounds [180]. Genotoxicity tests were also carried out on prokaryotic systems (bacteria). In contrast to the acellular assays, chlorogenic acid is non-mutagenic in bacterial mutagenicity assays. Tests on *Salmonella typhimurium* with and without metabolic activation on different strains were carried out, with different chlorogenic acid concentrations being examined [182,183,184,185,186]. One reason for the lack of mutagenic activity in tests with prokaryotes is probably because, unlike isolated DNA, they have DNA repair mechanisms. The sole consideration of mutagenicity tests at the molecular level on isolated DNA is interesting, but living beings have intact cells and organs that have the appropriate repair mechanisms and can repair occurring DNA strand breaks, gene mutations, and/or cytotoxic damage. In contrast, in lower eukaryotic systems such as *Saccharomyces cerevisiae* (strain D7), chlorogenic acid at a concentration of 3 mM (1 mg/mL) induced mitotic gene conversion in alkaline medium at pH 10 without metabolic activation (absence of S9 mix) [187].

However, in vitro assays with polyphenols like chlorogenic acid show controversial results. Wood et al. reported that chlorogenic acid (500 μM; 177 μg/mL) did not induce 8-azaguanine resistance in Chinese hamster V79-6 cells (in the absence of S9 activation), and the observed dose was not cytotoxic [188]. Fung et al. achieved a positive result with Chinese hamster V79 cells with a dose of 0.07 mg/mL chlorogenic acid. Chlorogenic acid was also mutagenic when tested with mouse lymphoma L5178Y cells but was negative in the absence of S9 mix (without metabolic activation) [185]. In vitro studies have shown that chlorogenic acid has clastogenic effects on mammalian cells. Treatment with chlorogenic acid resulted in the induction of chromosomal aberrations in Chinese hamster ovary (CHO) cells, as observed in studies conducted by Whitehead et al. and Stich et al., even in the absence of S9 activation enzymes [184,190]. However, it should be noted that the addition of S9 activation enzymes eliminated the clastogenic activity of chlorogenic acid. Unlike the clastogenic effects observed in vitro for chlorogenic acid, in vivo studies showed no induction of chromosomal damage. When male Sprague-Dawley rats were administered two oral doses of chlorogenic acid at 150 mg/kg (420 μmol/kg) bw with a 24 h interval, there were no increases in the frequencies of micronucleated polychromatic or normochromatic erythrocytes in the bone marrow, as reported by Hossain et al. in 1976 [192]. A recent intraperitoneal study by Alarcón-Herrera et al. (2017) in mice with 100 mg/kg CQA also showed no visible signs of genotoxicity/mutagenicity and thus confirmed the non-mutagenic effect of CQA [193]. In silico toxicity model calculations conducted in current research predict that both 5-caffeoylquinic acid and 3,5-dicaffeoylquinic acid are non-mutagenic (*p* = 0.93 and *p* = 0.85) [194], and there are even numerous studies in which chlorogenic acids are linked to antigenotoxic properties [195,196,197,198]. On the other hand, there are no known studies on the genotoxic/mutagenic potential of 3,5-DCQA. However, the cytotoxicity of 3,5-DCQA on human erythrocytes was evaluated by a hemolytic assay, which confirmed that 3,5-DCQA exhibited a low cytotoxicity against human erythrocytes [199].

### 6.3. Carcinogenicity, Reproductive Toxicity, and Teratogenic Effects

In 1976, Chaube et al. conducted an intraperitoneal study on pregnant female Wistar rats and reported that chlorogenic acid (5–500 mg/kg per day, total of 8 injections) was non-toxic and had no effects on the reproduction of the rats [173]. Furthermore, the authors suggested that CQA did not induce fetal central nervous system defects or maternal or fetal mortality. Three publications even indicate that CQA has a protective effect on reproductive toxicity. In a study by Mentese et al., in thirty Sprague-Dawley rats, 5-fluorouracil-induced toxicity was significantly reversed with CQA administration in a dose-dependent manner, and CQA acted as a modulator in attenuating xenobiotic-induced ovotoxicity [200]. In another publication, scientists were able to show that CQA counteracted arsenic-induced testicular dysfunction in adult male Swiss mice [201]. In an in vivo study in mice, CQA significantly suppressed zearalenone-induced ovarian granulosa cell death at doses ranging from 250 to 1000 µg/mL [202]. Carcinogenicity studies have also been carried out. In a study with 49 Swiss albino mice, CQA in purified cholesterol pellets was introduced into the urinary bladder and the mice were observed for one year [203]. Surprisingly, no bladder carcinomas could be found in the mice, from which it can be deduced that CQA is not carcinogenic. This was also proven by scientists in a study with male and female Syrian golden hamsters administered 0.025% CQA in the diet over 24 weeks, where CQA did not induce liver or large intestine tumors [198]. In another study, Swiss mice received CQA (25 mg/kg) intragastrically five times per week over a period of 10 weeks [204]. It could be shown that CQA attenuated early-stage colorectal carcinogenesis induced by 1,2-dimethylhydrazine/deoxycholic acid. To the best of the authors’ knowledge, no studies on the carcinogenicity of 3,5-DCQA are known; however, using an in silico computation prediction model [194], 3,5-DCQA was found to be non-carcinogenic (*p* = 0.63).

### 6.4. Neurotoxicity

Neurotoxicity describes any adverse effect on the nervous system (e.g., paralysis or loss of function) resulting from exposure to potentially toxic substances. A search of the literature did not find any evidence of neurotoxic effects of chlorogenic acids. On the contrary, scientific studies even show that chlorogenic acids and isochlorogenic acids have neuroprotective effects. Mikami et al. reported on a neuroprotective effect of 5-CQA in glutamate-induced neuronal cell death using primary cultures of mouse cerebral cortex [205]. Herein, CQA prevented the increase in intracellular concentration of Ca^2+^ caused by the addition of glutamate and protected neurons by regulating Ca^2+^ entry.

Aluminum compounds are also potent neurotoxins that cause oxidative stress and cognitive damage and have shown to be associated with Alzheimer’s disease. In experiments by Wang and colleagues on aluminum-induced neurotoxicity in mice, the chronic administration of CQA (i.g.) in doses of 50–200 mg/kg bw (aluminum chloride (AlCl_3_), 35 mg/kg bw per day) weakens aluminum-induced Al^3+^-accumulation, oxidative stress, mitochondrial damage, and nuclear pyknosis in the hippocampus [206,207]. Extracts containing 5-CQA, quercetin and other polyphenols have also been used successfully against the neuromuscular blockage caused by the venoms of *Bothrops jararacussu* (pit viper) and *Dabioa russelii* (Russell’s viper) in in vitro studies [208,209]. To obtain detailed information about the signaling pathway, Yao et al. induced oxidative damage in rat pheochromocytoma cells using hydrogen peroxide (H_2_O_2_) [210]. Herein, 5-caffeoylquinic acid showed robust free-radical-scavenging activity in vitro and rescued cells from oxidative insults by activation of the transcription factor Nrf2. Furthermore, it was found that chlorogenic acid increased the recovery of synaptic transmission upon reoxygenation in an in vitro ischemia model after exposure to β-amyloid peptide 1–42 (2 nmol, administered intracerebroventricularly (i.c.v.)) [211].

Several in vivo studies have demonstrated the neuroprotective properties of 5-CQA. These studies showed that 5-CQA can improve the pathological damage of hippocampal neurons, reduce neuronal death and apoptosis, alleviate epilepsy-like seizures and cognitive impairment, and ameliorate hippocampal neuronal degeneration and neuroinflammation [212,213,214,215,216]. Rebai and colleagues also found that 5-CQA has the potential to protect against AMPA-mediated excitotoxicity and may be a promising candidate for the prevention of neurodegenerative disorders associated with the loss and damage of oligodendrocytes, which play a major role in the myelination of axons [217]. Furthermore, another publication revealed the neuroprotective effect of DCQAs. Specifically, the study showed that 3,5-DCQA (as well as 4,5-DCQA and 1,5-DCQA) can alleviate stress-hormone-induced depressive behavior, including memory loss, in corticosterone-treated ICR mice by reducing reactive oxygen species (ROS) and inhibiting the activity of monoamine oxidase (MAO) type A and B in neurons and astrocytes. The study also discovered that 3,5-DCQA can protect against neuronal/dendritic atrophy and synaptic glutamatergic transmission in the hippocampus [218]. In summary, 5-CQA and 3,5-DCQA are not neurotoxic, and they even have possible neuroprotective properties.

### 6.5. Immunotoxicity and Allergenicity

As early as 1961, Freedman and colleagues reported that CQA was found to be an important allergenic constituent in green coffee beans, castor beans and oranges, and that coffee workers were occupationally exposed to finely dispersed coffee dust, which caused asthma, rhinitis, and dermatitis [219]. Additional studies in coffee workers showed positive wheal and erythema responses to intradermal injections [220,221,222,223,224]. The aim of those studies was also to investigate the immunotoxic potential of emulsions of CQA (1 mg/mL in physiological saline) in albino rabbits and guinea pigs immunized by the intravenous route. After six hours, the animals showed an immunological response, with antibody formation and evidence of erythema and induration. No severe central necrosis of the lesion was visible after twelve hours, and the intradermal response to CQA was Arthus-type.

The administration of 1 mg/mL CQA to guinea pigs by intracardiac injections causes death after 20–25 min (asphyxia), which is evidence of active systemic anaphylaxis, and CQA also induced complete hemagglutination. Although Freedman et al. found that CQA in plant materials appeared to cause allergic reactions in rabbits and humans, particularly when administered by injection, Layton et al. believed that pure CQA was not an allergen and that the allergic reactions described by Freedman and co-workers were mediated by the proteins in plant materials (as impurities or contaminants from the moiety of the seed) [225,226]. They concluded that CQA does not show any allergenic activity in clinical tests and is therefore to be rated as nonallergenic. This led to a discrepancy between the results of the two research groups and thereafter caused some public discussions [227]. More than 20 years later, De Zotti et al. reported that 14% of occupationally exposed coffee workers complained of allergenic symptoms in the eye, nose, and bronchial system [228]. The sensitization of workers (10%) was confirmed by the skin prick test, and they concluded that eliminating environmental dust during shipping operations was the most important preventive measure.

Recent studies indicate that CQA may not pose a safety risk to humans, as it does not elicit immunoreactivity even after a single subcutaneous exposure, as demonstrated by the reporter antigen popliteal lymph node assay (RA-PLNA) in BALB/c mice [229]. In addition, Li et al. cast doubt on Freedman’s findings, stating that “the subcutaneous administration of CQA into the footpad deviates from the normal routes of exposure, and drug or chemical metabolism in vivo may impact its bioavailability” [230]. In 2012, Lin and co-workers found that both 5-CQA and 3,5-DCQA significantly enhanced the secretion of trinitrophenyl ovalbumin-specific immunoglobulin IgG in an intravenous exposure mouse model [231]. The results suggest that the strength and type of immune effects of the sensitization response may correlate with structural differences in the CQA family, and the scientists were able to confirm the immunostimulatory activity of CQA by QSAR analysis in silico. Interestingly, the immunotoxicity for both 5-CQA and 3,5-DCQA was also predicted to be positive (*p* = 0.99) using the in silico tool from Banerjee et al. [194]. In apples, Unterhauser et al. recently demonstrated that the interaction of CQA with proteins (formation of adducts as a hapten–protein conjugate) via reactive *o*-benzoquinone in vitro causes a loss of conformational epitopes and a decrease in the IgE-binding capacity by the direct reaction of CQA with the cysteine moiety of IgE [232]. Interestingly, the high content of CQA in apples results in a lower allergenic potential. Other scientific research groups have observed the covalent interaction of CQA with proteins as well, and they also found low allergenic activity [224,233,234].

There has been a long-standing debate regarding whether CQA exhibits antigenic and allergenic activities, with two opposing views. However, as per drug metabolism theories, small-molecule drugs can initiate an immune response by forming stable adducts with proteins. Such protein adducts can form through direct chemical reactions or by producing electrophilic metabolites, acting as an antigenic signal and triggering an immune response.

Based on the publications cited here, and to the best of the authors’ knowledge, it can be concluded that CQA and 3,5-DCQA in their pure form (without interaction with proteins) are likely to have a low immunotoxic effect, while the direct interaction of CQA with proteins is likely to induce a greater immune response. However, further studies on the effects of isolated chlorogenic acid should be undertaken to conclusively determine whether CQA has effects on the immune system.

### 6.6. Other Adverse Effects

Chlorogenic acid has been found to counteract the effects of metformin, a pharmaceutical drug used to manage elevated blood sugar levels. This interaction is only observed at high levels of chlorogenic acid, which is unlikely to occur in humans and thus may not have a significant impact [235,236]. So far, no other adverse effects for 5-CQA and/or 3,5-DCQA have been documented.

## 7. Regulatory and Nutritional Information

Currently, due to a lack of data, the EFSA has not yet derived any health-based guidance values, such as tolerable or acceptable daily intake levels (TDI/ADI), for both 5-caffeoylquinic acid and 3,5-dicaffeoylquinic acid. Thus, to the best of the authors’ knowledge, there are currently no specific regulations in place in the EU or other regions worldwide. Nevertheless, an application to authorize the placing on the market of an infusion from the coffee leaves of *Coffea arabica* and/or *Coffea canephora* as a traditional food from a non-EU country under Regulation (EU) 2015/2283 of the European Parliament and of the Council has been approved, and the EU Commission has stated a maximum permitted level of 5-CQA of <100 mg/L in tea-like infusions prepared with 20 g of dried leaves in 1 L of hot water [237].

## 8. Exposure and Risk Assessment

While the direct consumption of large amounts of coffee by-products is unlikely, it is important to recognize the various ways in which individuals can be exposed to chlorogenic and isochlorogenic acids. These compounds can enter the body through a variety of sources, including food, health products, medications, or processed products that contain coffee by-products as ingredients [4,10,15,22]. For example, coffee by-products may be used in the production of dietary supplements, functional foods, or pharmaceutical formulations [35,51,118]. In addition, coffee by-products can be processed and extracted to produce concentrated forms of chlorogenic and isochlorogenic acids for use in the food and beverage industry [15]. By considering these additional routes of intake, the following risk assessment aims to provide a comprehensive understanding of potential exposure scenarios and to make the scope of the evaluation as broad as possible.

### 8.1. Theoretical Maximum Daily Intake (TMDI) and Limitations

In order to determine the oral exposure of the general population in the EU to 5-CQA and 3,5-DCQA from coffee by-product beverages, information on daily consumption is needed. In the absence of representative data on the daily per capita consumption of coffee by-products in the EU, a realistic worst-case scenario was assumed.

It is known that tea-like infusions can be made from coffee by-products [4,10,15]. Recently, the EU Commission approved the placing on the market of an infusion made from the coffee leaves of *C. arabica* and/or *C. canephora* according to Regulation (EU) 2015/2283 [11,237]. The consumption of coffee by-products as an aqueous infusion is therefore considered a plausible scenario. The following assumptions were made for considering the worst-case scenario: a single serving is prepared with 2 g of the relevant coffee by-product and 200 mL of cold water, in accordance with the method of Steger et al. and the application on the risk assessment of other ingredients in coffee and coffee by-products [238,239]. This assumption was made because chlorogenic acids and isochlorogenic acids are sensitive to the influence of pH and temperature and can undergo degradation and conversion reactions that could significantly affect the actual levels of 5-CQA and 3,5-DCQA (as already described in Section 3.1). Furthermore, 5-CQA and 3,5-DCQA are sufficiently soluble in water that full extraction from coffee by-products can be expected (see Section 5 and Table 3).

From a large epidemiological study in Germany (“Nationale Verzehrsstudie II” from Max-Rubner Institute (MRI)) with about 19,000 participants (~9000 males, ~10,000 females, aged from 14 to 80 years), representative data on dietary consumer habits and daily food consumption of tea-like infusions could be obtained [240]. In the category “alcohol-free beverages” and the subgroup “herbal/fruity tea”, the results of the study (95th percentile) show that women consume 1300 g of herbal/fruity tea daily (in comparison, males only consume 800 g/day). Therefore, a maximum consumption of 1300 g is considered as the worst-case scenario. To simplify the calculation of the Theoretical Maximum Daily Intake (TMDI), a density of 1000 kg/m^3^ (from water) for an aqueous infusion is assumed, so that 1300 g is equivalent to 1300 mL. Table 5 shows the calculated TMDIs of the coffee by-products, bearing in mind that coffee parchment and spent grounds are not yet commonly used alimentarily, such as in tea-like infusions.

The results show that the TMDI of coffee husks and green unroasted beans for 5-CQA is 1.74 g/day, the highest among all coffee by-products. For 3,5-DCQA, however, the TMDI of coffee flowers (blossoms) has the highest value at 0.76 g/day. The TMDIs of 5-CQA and 3,5-DCQA in coffee parchment and cascara are very small in comparison and can therefore be ignored. In summary, the TMDI of 5-CQA for all coffee by-products is 0.71 g/day, whereas the total TMDI of 3,5-DCQA is 0.19 g/day, which is 3.7 times lower than that of 5-CQA. In relation to coffee, containing a maximum of 9.0 g/100 g 5-CQA (see Section 4 and [114,115]), the content per serving results in 0.18 g/200 mL and in a theoretical maximum daily intake of 1.17 g/day.

### 8.2. Acute Oral Exposure

For the acute oral exposure assessment of 5-CQA and 3,5-DCQA from coffee by-products, only the experimentally established LD_50_ (p.o.) values were used as a toxicological threshold. This procedure was introduced by Gold and co-workers [241] as a recognized procedure, in which the LD_50_ values were extrapolated to the lower 99% one-sided confidence limit of the benchmark dose values (BMDL_10_). In this simple and quick estimate, the BDML_10_ can be obtained when the LD_50_ value is divided by a factor of 10.2, assuming a linear dose–response relationship and no other dose–response information is available. Various LD_50_ values were determined for 5-CQA (see Section 6.1). An LD_50_ value of 2000 mg/kg bw is assumed to determine the BMDL_10_, which was carried out in experiments on rats and mice [167,168]. The higher LD_50_ value of 4000 mg/kg bw from the PubChem database [140] was not used to determine the acute oral exposure, as this value was determined in rats by intraperitoneal administration (and not by oral treatment). The LD_50_ of 2000 mg/kg bw results in a BMDL_10_ of 196 mg/kg bw for 5-CQA. For an average person who weighs 70 kg, the extrapolated BMDL_10_ corresponds to an intake of 13.7 g of 5-CQA per day. Interestingly, Costa Silva Faria et al. determined an LD_100_ of 5000 mg/kg bw and derived a NOAEL of 1000 mg/kg/day for mice, which corresponds to a NOAEL of 13.2 g/day for a 70 kg human adult. By comparing the BMDL_10_ (13.7 g/day) with the NOAEL (13.2 g/day), it is noticeable that both values are very similar. There are also different lethal doses for 3,5-dicaffeoylquinic acid. For safety reasons, the lowest LD_50_ value of 2154 mg/kg bw resulting from a study with rats by oral administration was used [176]. For 3,5-DCQA, a BMDL_10_ result of 211 mg/kg bw is in accordance with Gold and colleagues [241], which is equivalent to an intake of 14.8 g of 3,5-DCQA per day for an average person weighing 70 kg.

The results in Table 6 show that humans would have to consume unlikely amounts of tea-like infusions of coffee by-products per day to induce a toxic effect. Such calculated values are unrealistic (i.e., even well above excessive total daily fluid intake) and beyond any amount of coffee by-products that could be expected to be consumed by an individual in a day. Therefore, no adverse or toxic effects are expected from acute oral exposure to 5-CQA or 3,5-DCQA from the consumption of coffee by-products.

### 8.3. Chronic Oral Exposure

People who consume 5-caffeoylquinic acid (5-CQA) or 3,5-dicaffeoylquinic acid (3,5-DCQA) daily through the consumption of coffee by-products are chronically exposed. Sub-chronic studies showed that 5-CQA and 3,5-DCQA demonstrated no significant adverse effects at doses within the normal range of dietary exposure. The logP value and good water solubility (see Table 3), and metabolism and the pharmacokinetic parameters such as the volume of distribution (V_D_) and clearance indicate (see Section 5) that both the main representative of the monocaffeoylquinic acids, 5-CQA, and also the main representative of dicaffeoylquinic acids, 3,5-DCQA, cannot be accumulated in the human body. Forestomach hyperplasia was only observed in one study on rats [171,172], but it is pathologically harmless and disappears in the absence of the stimulus. No CQA-induced toxic or adverse effects have been observed in animals or humans in studies on carcinogenicity, reproductive toxicity, teratogenic effects, and neurotoxicity. Conducted in vivo and in vitro genotoxicity tests were also negative, which is why 5-CQA and 3,5-DCQA are classified as non-genotoxic. CQA-induced DNA mutations in genotoxicity tests were observed on isolated DNA, but tests on isolated DNA are not informative for risk assessment because they do not take into account any type of repair mechanisms that exist in living beings. Studies showed ambivalent results for the immunotoxicity of 5-CQA and 3,5-DCQA (see Section 6.5). A few studies reported allergenic reactions, particularly in occupational exposed workers. While chlorogenic acids are natural plant compounds found in a variety of foods, including fruits, vegetables, coffee and coffee by-products, for individuals with a known allergy to one or more of these foods, exposure to chlorogenic acids may trigger an allergic reaction. Further research is needed to determine whether chlorogenic and isochlorogenic acids are immunotoxic to the human population. When chlorogenic acids are consumed with food in normal amounts as a component of coffee by-products, no adverse health effects from this class of compounds appear to be possible (see Table 6).

## 9. Conclusions

The present review evaluated the potential health risks associated with the oral consumption of chlorogenic and isochlorogenic acids from coffee by-products, based on current pharmacokinetic and toxicological knowledge and the available consumption data for the general population in the EU. No significant signs of toxicity or adverse effects were observed after acute oral exposure. Based on current knowledge, long-term exposure to chlorogenic and isochlorogenic acids at levels typically found in coffee by-products does not appear to pose a health risk to humans. Consumption of CQAs from coffee by-products can therefore be considered safe based on the currently available literature. Due to the lack of more recent literature, including some of the issues noted above, the authors acknowledge that additional research is needed, particularly if exposure to chlorogenic acids or coffee by-products is expanded beyond currently expected levels.

## Figures and Tables

**Figure 1 molecules-28-05540-f001:**
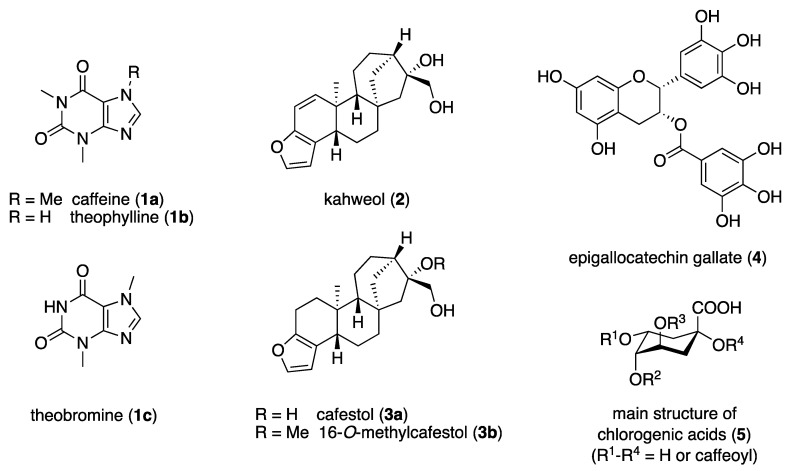
Major constituents of *Coffea* spp.

**Figure 2 molecules-28-05540-f002:**
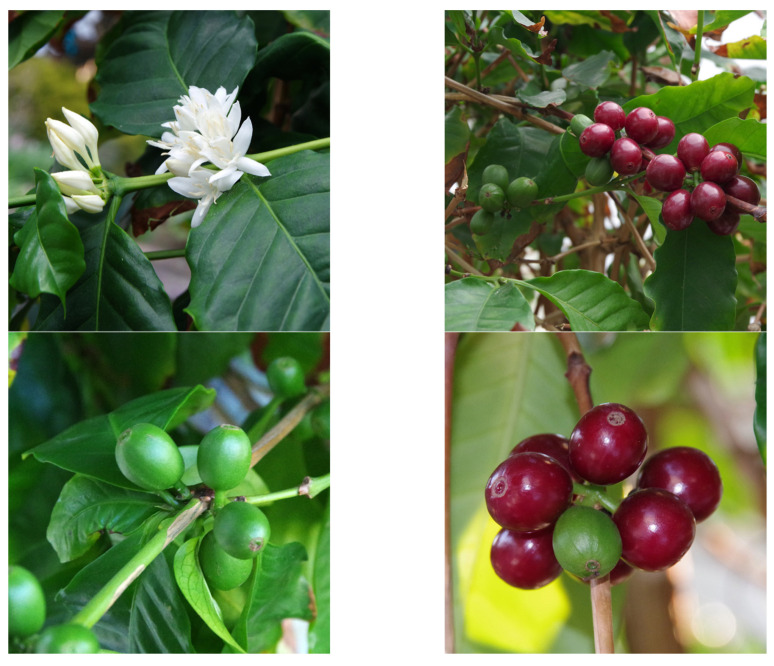
Some coffee by-products (coffee flowers/blossoms, coffee leaves, ripe and unripe (green) coffee cherries).

**Figure 3 molecules-28-05540-f003:**
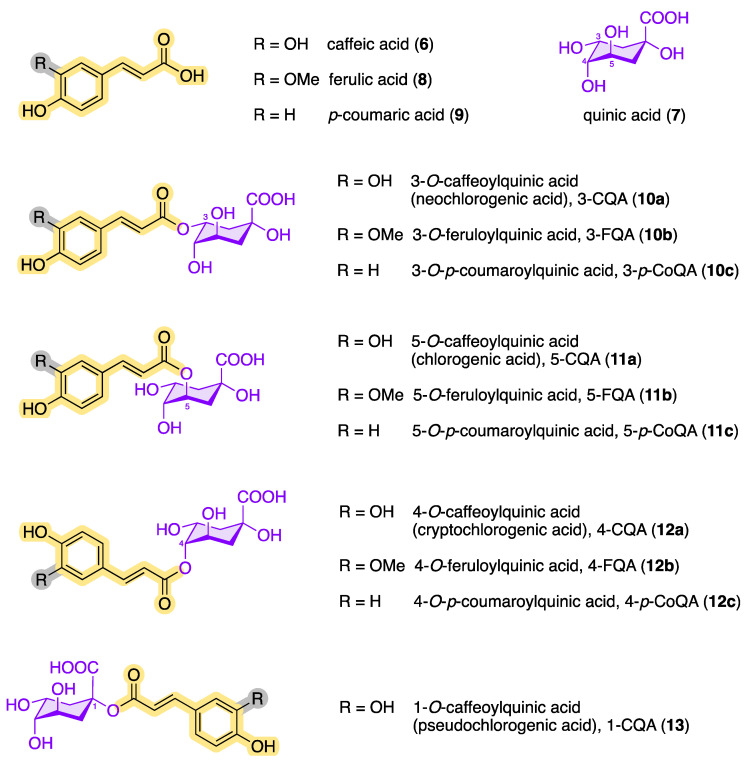
Chemical structures of quinic acid, hydroxycinnamic acids, and the most common naturally occurring CQAs (following the IUPAC numbering system).

**Figure 4 molecules-28-05540-f004:**
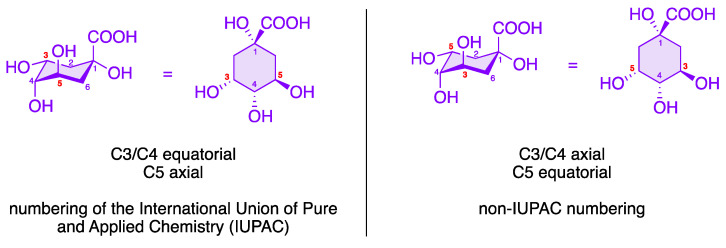
Numbering of the (–)-quinic acid fragment in CQAs.

**Figure 5 molecules-28-05540-f005:**
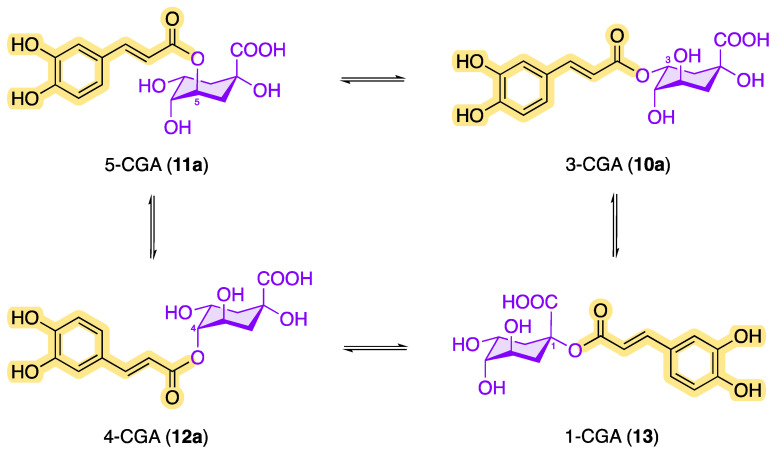
Conversion of CQAs by transesterification.

**Figure 6 molecules-28-05540-f006:**
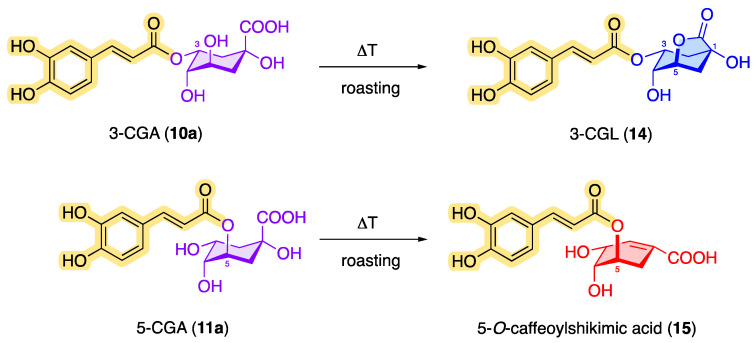
CQA products in roasted coffee: formation of 3-*O*-caffeoylquinic-1,5-γ-lactone (**14**) and 5-*O*-caffeoylshikimic acid (**15**).

**Figure 7 molecules-28-05540-f007:**
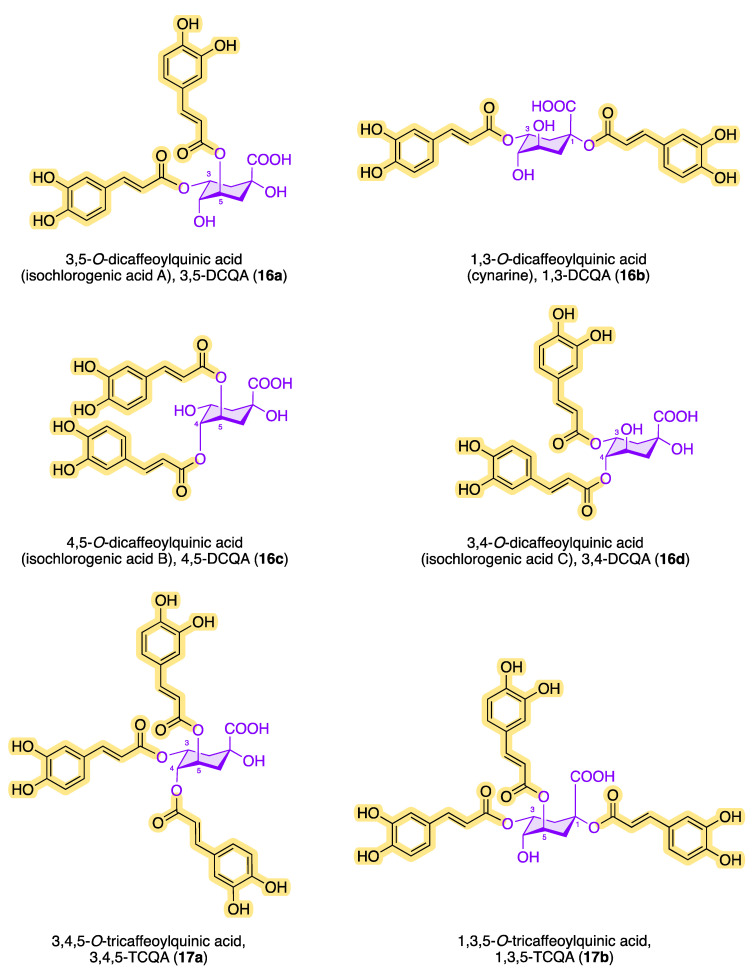
Structures of di-, tri- and tetra-caffeoylquinic acids.

**Figure 8 molecules-28-05540-f008:**
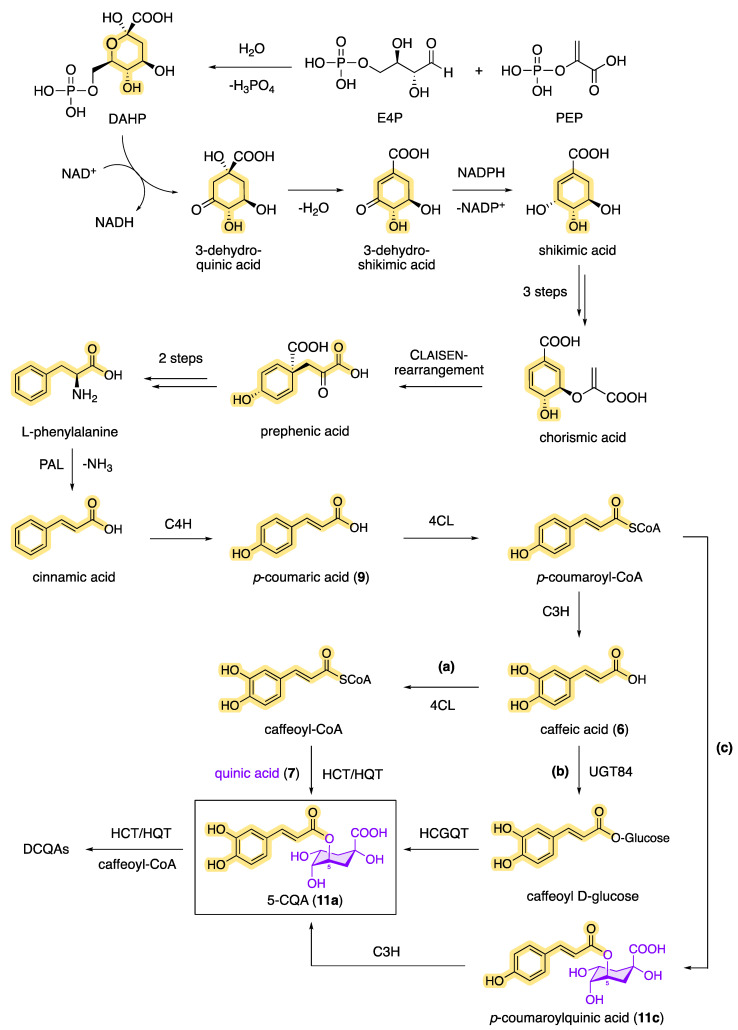
Biosynthetic pathway of chlorogenic acids (modified from [22,28,88,93]).

**Figure 9 molecules-28-05540-f009:**
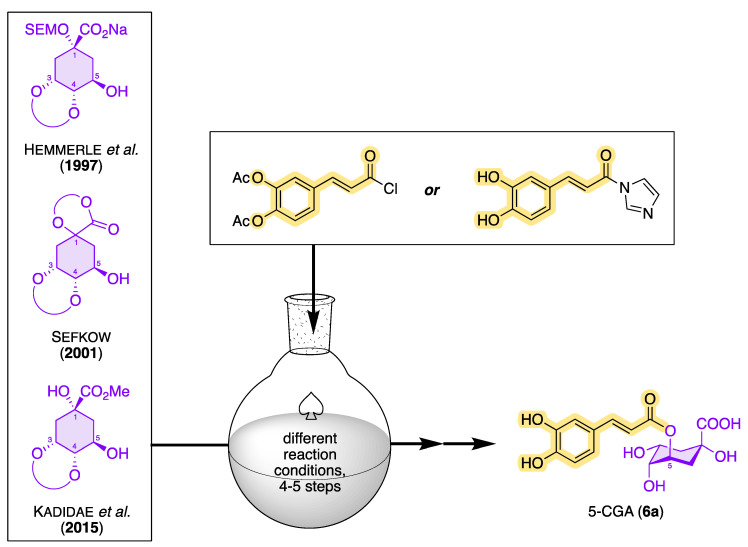
Approaches to the total synthesis of 5-CQA (examples) [100,101,102].

**Table 1 molecules-28-05540-t001:** Amounts of chlorogenic acids in coffee by-products.

Coffee By-Product	Chlorogenic Acid Content, Expressed in [g/100 g] (unless Otherwise Stated) *						Ref.
	3-CQA (10a)	4-CQA (12a)	5-CQA (11a)	3,4-DCQA (16d)	3,5-DCQA (16a)	4,5-DCQA (16c)	
coffee flowers (blossoms)	n.r.	n.r.	0.13–2.64	0.01–0.25	0.19–5.84	n.r.	[118]
	0.01–0.07 (in total)						[119]
coffee leaves	0.04–0.30	0.09–0.84	1.39–5.58	0.03–0.18	0.01–1.13	0.01–0.12	[120]
	1.1–18.0 mg/L	4.2–30.0 mg/L	1.0–190.0 mg/L **	n.r.	n.r.	n.r.	[121]
coffee pulp	n.r.	n.r.	0.03–0.13	n.r.	n.r.	n.r.	[122,123]
	n.r.	n.r.	2.3	n.r.	n.r.	n.r.	[124]
	0.03	0.04	0.22	0.05	0.06	0.03	[125]
	n.r.	n.r.	0.13–2.44	n.r.	n.r.	n.r.	[126]
	n.r.	n.r.	2.40	n.r.	n.r.	n.r.	[127]
coffee husk	n.r.	n.r.	13.3	n.r.	n.r.	n.r.	[124]
	0.01	0.02	0.57	0.08	0.15	0.04	[125]
	n.r.	n.r.	2.50	n.r.	n.r.	n.r.	[127]
	n.r.	n.r.	0.04–0.17	n.r.	n.r.	n.r.	[128]
	n.r.	n.r.	0.17	n.r.	n.r.	n.r.	[129]
	n.r.	n.r.	0.02–0.17	n.r.	n.r.	n.r.	[128]
	n.r.	n.r.	69.9 mg/L **	n.r.	n.r.	n.r.	[130]
silver skin	1.80–5.16	2.38–6.02	7.33–8.98	0.93–1.04	0.55–1.00	0.79–1.06	[131]
	0.15	0.09	0.20	n.r.	n.r.	n.r.	[132]
	n.r.	n.r.	2.2	n.r.	n.r.	n.r.	[133]
	n.r.	n.r.	0.94–2.13	n.r.	n.r.	n.r.	[129]
	n.r.	n.r.	0.39	n.r.	n.r.	n.r.	[134]
	n.r.	n.r.	20–30 mg/L **	n.r.	n.r.	n.r.	[130]
	n.r.	n.r.	0.02–0.04	n.r.	n.r.	n.r.	[134]
	n.r.	n.r.	3.00	n.r.	n.r.	n.r.	[127]
parchment	n.r.	n.r.	0.61	n.r.	n.r.	n.r.	[129]
	n.r.	n.r.	0.40	n.r.	n.r.	n.r.	[135]
spent coffee grounds	1.50	1.62	1.87	0.29	1.00	1.70	[131]
	n.r.	n.r.	2.30	n.r.	n.r.	n.r.	[127]
	0.62–1.32 (in total)			3.31–5.79 (in total)			[136]
green unroasted beans	1.25	1.94	13.4	0.66	1.10	0.63	[131]
	n.r.	n.r.	3.6–4.4	n.r.	n.r.	n.r.	[135]

* n.r. = not reported; ** data reported for a beverage prepared from the material. For standardization and for a better comparison of the values among each other, the values found in the literature were converted into the unit [g/100 g].

**Table 2 molecules-28-05540-t002:** Amounts of 5-caffeoylquinic acid in coffee by-products.

Coffee By-Product	Min (%)	Max (%)	Average (%)
coffee flowers (blossoms)	0.01	2.60	1.31
coffee leaves	1.39	5.60	3.50
coffee pulp	0.03	2.40	1.22
coffee husk	0.02	13.3	6.66
cascara	0.01	0.01	0.01
silver skin	0.02	8.98	4.50
parchment	0.40	0.61	0.51
spent coffee grounds	1.32	2.30	1.81
green unroasted beans	3.60	13.4	8.50
total	0.76	5.47	3.11

**Table 3 molecules-28-05540-t003:** Chemical and physical properties of 5-CQA and 3,5-DCQA [140] *.

Parameter	5-Caffeoylquinic Acid (5-CQA)	3,5-Dicaffeoylquinic Acid (3,5-DCQA)
CAS	327-97-9	2450-53-5
chemical formula	C_16_H_18_O_9_	C_25_H_24_O_12_
molecular weight	354.31 g/mol	516.45 g/mol
solubility	soluble in hot water (40 mg/mL at 25 °C), ethanol and acetone	soluble in DMSO (slightly) and methanol (slightly)
form	solid, white crystals	hygroscopic solid, off-white to pale yellow
pK_a_	3.33 [141]	3.54 (predicted)
logP	−0.4	1.0
MP	210 °C	170–172 °C
LD_50_	4000 mg/kg (rat, i.p.)	2154 mg/kg (rat, p.o.) [142]

* Data from the National Library of Medicine (NIH) at PubChem and MSDS from Merck (unless otherwise stated).

**Table 4 molecules-28-05540-t004:** Results for CQA in different genotoxicity and mutagenicity test systems (from summary of [179] with reference to the original literature and with the inclusion of further studies).

Test System	Strains	Metabolic Activation with S9 Mix	Dose of CQA	Result	Ref.
isolated λ DNA	-	-	250 µM (89 µg/mL)	positive	[179]
isolated plasmid DNA (pBR322)	-	-	100 µM (35 µg/mL)	positive (in the presence of NO-releasing compounds)	[180]
phage DNA (øX174 RF I)	-	-	n.p.	positive (in the presence of Cu^2+^)	[181]
*S. typhimurium* assay	TA98	+/−	0.17 or 1.7 µmol/plate (58.8 or 588 µg/plate)	negative	[182]
*S. typhimurium* assay	TA98	+	1, 3, 6 or 9 mg/mL (3, 9, 20 or 30 mM)	negative	[183]
*S. typhimurium* assay	TA98, TA100	+/−	19 or 28 mg/plate (53 or 79 µmol/plate)	positive (in the presence of Mn^2+^); negative (in the presence and absence of S9 mix and in the presence of Cu^2+^ only)	[184]
*S. typhimurium* assay	TA98, TA100, TA1535, TA1537, TA1538	+/−	0.33–10 mg/plate (0.94–28 µmol/plate)	negative	[185]
*S. typhimurium* assay	BA13	*-*	0.3–28 µmol/plate (0.1–9.9 mg/plate)	weakly positive	[186]
*S. cerevisiae* assay	D7	+/−	20, 40 or 80 mg/mL (56, 110 or 230 mM)	positive (w/o S9 mix); negative (w/S9 mix)	[184]
*S. cerevisiae* assay	D7	-	1 mg/mL (3 mM)	positive	[187]
Chinese hamster V79-6 cells	-	-	500 nmol/mL (177 µg/mL)	negative	[188]
Chinese hamster V79 cells	-	-	0.07 mg/mL	positive	[189]
Mouse lymphoma L5178Y cells	-	+/−	6.5–10 mg/mL (18.5–28 mM)	positive (w/S9 mix); negative (w/o S9 mix)	[185]
CHO cells (chromosomal aberration test)	-	+/−	10–40 µg/mL (29–110 µM)	positive (w/o S9 mix, but in the presence of Mn^2+^ and Cu^2+^); negative (w/S9 mix)	[184]
CHO cells (chromosomal aberration test)	-	+/−	125, 150 or 250 µg/mL (353, 420 or 706 µM)	positive (w/o S9 mix); negative (w/S9 mix)	[190]
HL-60 or Jurkat cells	-	n.a.	1–100 µM	negative	[191]
Micronucleus test in bone marrow (rats)	-	n.a.	150 mg/kg (420 µmol/kg, p.o., 24 h)	negative	[192]
Micronucleus test in bone marrow (mice)	-	n.a.	100 mg/kg (i.p.)	negative	[193]

Abbreviations: n.p. = not provided; n.a. = not applicable; w/ = with or “+”; w/o = without or “−”.

**Table 5 molecules-28-05540-t005:** Theoretical maximum daily intake (TMDI) of 5-CQA and 3,5-DCQA from coffee by-products via tea-like aqueous infusions.

Coffee By-Product	Consumption of Infusions (mL/day)	5-CQA		3,5-DCQA	
		Content Per Serving (g/200 mL) *	TMDI (g/day)	Content Per Serving (g/200 mL) *	TMDI (g/day)
coffee flowers (blossoms)	1300	0.052	0.338	0.117	0.759
coffee leaves	1300	0.112	0.728	0.023	0.147
coffee pulp	1300	0.048	0.312	0.001	0.008
coffee husk	1300	0.266	1.729	0.003	0.020
Cascara beverage	1300	0.000	0.001	n.d.	n.d.
silver skin	1300	0.180	1.167	0.020	0.130
coffee parchment	1300	0.012	0.079	n.d.	n.d.
spent coffee grounds	1300	0.046	0.299	0.020	0.130
green unroasted beans	1300	0.268	1.742	0.022	0.143
total average		0.109	0.711	0.029	0.191

* The content per serving in g/200 mL was calculated using the maximum amount of 5-CQA and 3,5-DCQA, respectively, found in coffee by-products from Table 1; n.d. = no data.

**Table 6 molecules-28-05540-t006:** Beverage volume (in L) for reaching the oral BMDL_10_ for 5-CQA and 3,5-DCQA *.

Coffee By-Product	5-CQA	3,5-DCQA
	Beverage Volume (in L/day) for Reaching Oral BMDL_10_ of 196 mg/kg bw	Beverage Volume (in L/day) for Reaching Oral BMDL_10_ of 211 mg/kg bw
coffee flowers (blossoms)	53	25
coffee leaves	24	131
coffee pulp	57	2467
coffee husk	10	987
Cascara beverage	13,700	n.d.
silver skin	15	148
coffee parchment	225	n.d.
spent coffee grounds	60	148
green unroasted beans	10	135

* Estimation of BMDL_10_ obtained from LD_50_ values (2000 mg/kg bw for 5-CQA and 2154 mg/kg bw for 3,5-DCQA) using method B described by Gold et al. [241]; n.d. = no data.

## Data Availability

No new data were created or analyzed in this study. Data sharing is not applicable to this article.

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
