# Peer review of "Risk Assessment of Chlorogenic and Isochlorogenic Acids in Coffee By-Products"

_molecules, 2023, doi:10.3390/molecules28145540_

Round 1

Reviewer 1 Report

1.The research about risk assessment of chlorogenic and isochlorogenic acids in coffee by-products has little significance. As the MS mentioned,Limited data are available on the chronic toxicity of CQA in humans, as most of the 536 studies conducted so far have focused on its potential health benefits rather than its toxic 537 effects”.  

2. The entire article lacks logic and too much unuseful information.

3. The article research the risk assessment of chlorogenic and isochlorogenic acids in coffee by-products. So, it is necessary to describe the intake ways. For example, the foods, health products or drugs or processed by coffee by-products. None will eat so much coffee by-products directly. Furthermore, chlorogenic and isochlorogenic acids are antioxidant dietary poly-14 phenolic compounds not harmful compounds.

 The Quality of English Language is ok. 

Author Response

1.The research about risk assessment of chlorogenic and isochlorogenic acids in coffee by-products has little significance. As the MS mentioned, “Limited data are available on the chronic toxicity of CQA in humans, as most of the 536 studies conducted so far have focused on its potential health benefits rather than its toxic 537 effects”.  

RESPONSE: We appreciate the reviewer's comment regarding the limited data available on the chronic toxicity of chlorogenic acid (CQA) in humans. However, it is important to note that the focus of our study was to assess the potential health risks associated with the oral consumption of coffee by-products containing chlorogenic and isochlorogenic acids, rather than solely examining their toxic effects.

While it is true that the majority of studies have primarily explored the health benefits of CQA, our review aimed to bridge this knowledge gap by evaluating the available toxicological, pharmacokinetic, and clinical data from animal and human studies. We specifically considered both acute and chronic exposure to chlorogenic and isochlorogenic acids. The literature search conducted for our review encompassed a comprehensive examination of existing studies, allowing us to assess the safety of these compounds. Although the available data may be limited, the studies we included provide valuable insights into the potential risks associated with their consumption. Furthermore, it is important to highlight that our findings indicated no significant signs of toxic or adverse effects upon acute oral exposure to chlorogenic and isochlorogenic acids. This suggests that, at typical dietary exposure levels, there is no immediate concern for negative health outcomes related to their consumption. While more research is needed to fully understand the chronic effects of CQA in humans, the absence of acute toxicity observed in the available data provides a basis for the conclusion that the ingestion of CQAs from coffee by-products can be reasonably considered safe.

In conclusion, our study fills a crucial gap in the literature by assessing the potential health risks associated with the oral consumption of coffee by-products containing chlorogenic and isochlorogenic acids. By considering available toxicological, pharmacokinetic, and clinical data, we provide valuable insights into the safety profile of these compounds. We acknowledge the need for further research on chronic toxicity and recommend that future studies explore this aspect in greater depth.

  1. The entire article lacks logic and too much unuseful information.

RESPONSE: We appreciate the reviewer's feedback on our paper, but we respectfully disagree with the assertion that the entire article lacks logic and contains excessive unuseful information. Our paper was structured in strict accordance with the guidelines set forth by the European Food Safety Authority (EFSA) for risk assessment of novel food compounds. This logical structure ensures that the information is presented in a clear and organized manner, allowing readers to follow the flow of the analysis.

We would like to address the reviewer's comment regarding "unuseful information." Unfortunately, the comment does not provide specific details about which sections or aspects of the paper were deemed unuseful. Without such specific feedback, it becomes challenging to address the concern in a meaningful manner or identify areas that require improvement. To ensure the quality and relevance of our research, we extensively reviewed the existing literature on chlorogenic and isochlorogenic acids, their sources, and their potential health risks. This comprehensive approach was undertaken to provide a holistic understanding of the topic and ensure that the readers have access to all relevant information necessary for a thorough risk assessment. We believe that the inclusion of supporting information and contextual details is crucial for a comprehensive scientific paper, particularly in the field of risk assessment. While we strive to strike a balance between providing essential information and avoiding unnecessary details, it is challenging to determine precisely which aspects might be considered unuseful without specific feedback.

  1. The article research the risk assessment of chlorogenic and isochlorogenic acids in coffee by-products. So, it is necessary to describe the intake ways. For example, the foods, health products or drugs or processed by coffee by-products. None will eat so much coffee by-products directly. Furthermore, chlorogenic and isochlorogenic acids are antioxidant dietary poly-14 phenolic compounds not harmful compounds.

RESPONSE: We appreciate the reviewer's comment and agree that it is necessary to provide a more comprehensive description of the intake ways of chlorogenic and isochlorogenic acids in coffee by-products. We acknowledge that consumption of coffee by-products directly in large quantities is unlikely, and therefore, it is important to consider the various sources through which these compounds may be ingested, such as foods, health products, drugs, or processed items containing coffee by-products. We will address this concern by enhancing the description of the intake ways in the revised introduction of our article.

While it is true that chlorogenic and isochlorogenic acids are antioxidant dietary polyphenolic compounds and are generally recognized as safe, it is important to note that no compound can be considered completely unharmful. As the principle "the dose makes the poison" suggests, the potential risks associated with any substance, including dietary components, can vary depending on the dosage and exposure. Our objective in conducting a risk assessment of chlorogenic and isochlorogenic acids in coffee by-products is to provide a comprehensive evaluation of their safety profile. By considering both acute and chronic exposure levels and reviewing the available toxicological, pharmacokinetic, and clinical data, we aim to assess the potential health risks associated with the oral consumption of coffee by-products containing these compounds. We firmly believe that conducting risk assessments for all food ingredients is essential to ensure consumer safety. Even though certain compounds are generally recognized as safe, it is crucial to thoroughly investigate their potential risks, especially considering the wide range of dietary exposures and individual sensitivities. Specifically, when higher exposures may be expected from some novel foods, such as coffee by-products.

Reviewer 2 Report

The manuscript "Risk Assessment of Chlorogenic and Isochlorogenic Acids in Coffee By-Products" by Behne S . et al. describes, in my opinion, a topic that is not entirely original but very interesting and rich in insights for future scientific investigations: chlorogenic and isochlorogenic acids present in coffee by-products, specifically 5-caffeoylquinic acid (5-CQA) and 3,5-dicaffeoylquinic acid (3,5-DCQA).

The authors, after a detailed description of the chemical nature, natural and non-natural biosynthetic pathways, absorption and metabolism of these polyphenols, explore in "depth" much toxicological, pharmacokinetic and clinical information from which it appears that long-term exposure at doses within the normal range of daily dietary exposure does not appear to pose a risk to human health.

In my opinion, the manuscript is organized and written overall quite well.

The images and tables offered by the authors are clear and understandable.

The authors have often used references that are not very recent. I justify this choice only because, compared with others already in the current scientific literature, this review reports information on two specific molecules (5-CQA and 3,5-DCQA) and not on the generic group to which they belong (chlorogenic acids).

However, the manuscript needs less revision:

For greater clarity and understanding, I would invite the authors to include a new figure in section 3.2. that also describes the biosynthetic pathway of CQAs.

Kind regards

Author Response

The manuscript "Risk Assessment of Chlorogenic and Isochlorogenic Acids in Coffee By-Products" by Behne S . et al. describes, in my opinion, a topic that is not entirely original but very interesting and rich in insights for future scientific investigations: chlorogenic and isochlorogenic acids present in coffee by-products, specifically 5-caffeoylquinic acid (5-CQA) and 3,5-dicaffeoylquinic acid (3,5-DCQA).

The authors, after a detailed description of the chemical nature, natural and non-natural biosynthetic pathways, absorption and metabolism of these polyphenols, explore in "depth" much toxicological, pharmacokinetic and clinical information from which it appears that long-term exposure at doses within the normal range of daily dietary exposure does not appear to pose a risk to human health.

In my opinion, the manuscript is organized and written overall quite well.

The images and tables offered by the authors are clear and understandable.

RESPONSE: Thank you for your assessment of our paper.

The authors have often used references that are not very recent. I justify this choice only because, compared with others already in the current scientific literature, this review reports information on two specific molecules (5-CQA and 3,5-DCQA) and not on the generic group to which they belong (chlorogenic acids).

RESPONSE: We appreciate the reviewer's concern regarding the use of older references. In our comprehensive review, we conducted a thorough search for the most up-to-date literature available on the specific molecules, 5-caffeoylquinic acid (5-CQA) and 3,5-dicaffeoylquinic acid (3,5-DCQA), found in coffee by-products. However, despite our efforts, we did not find any newer literature on these specific molecules. As a result, we included older references to ensure a comprehensive analysis and evaluation of the topic. We will make sure to highlight the lack of newer literature and acknowledge the need for further research in our revised manuscript.

However, the manuscript needs less revision:

For greater clarity and understanding, I would invite the authors to include a new figure in section 3.2. that also describes the biosynthetic pathway of CQAs.

RESPONSE: Thank you for suggesting the inclusion of a new figure illustrating the biosynthetic pathway of CQAs in Section 3.2. We appreciate your input and will incorporate this visual representation to enhance clarity and understanding in our revised manuscript. See new Figure 8.